# Heat Shock Proteins in Gastrointestinal and Lung Neuroendocrine Neoplasm: Diagnostic and Therapeutic Perspectives

**DOI:** 10.3390/cells14191501

**Published:** 2025-09-25

**Authors:** Jacek Kabut, Jakub Sokołowski, Wiktoria Żelazna, Mateusz Stępień, Marta Strauchman, Natalia Jaworska, Anita Gorzelak-Magiera, Jakub Wnuk, Iwona Gisterek-Grocholska

**Affiliations:** 1Department of Oncology and Radiotherapy, Faculty of Medical Sciences in Zabrze, Medical University of Silesia in Katowice, 40-615 Katowice, Poland; jacekkabut@gmail.com (J.K.); jkb.wnuk@gmail.com (J.W.); igisterek@sum.edu.pl (I.G.-G.); 2Student Scientific Club at the Department and Clinic of Oncology and Radiotherapy, Faculty of Medical Sciences in Katowice, Medical University of Silesia in Katowice, 40-615 Katowice, Poland; czjaslo@gmail.com (J.S.); wikzel575@gmail.com (W.Ż.); matthew.x.stepien@gmail.com (M.S.); poczta.martys@gmail.com (M.S.); nataliajaworska@op.pl (N.J.)

**Keywords:** Heat Shock Proteins (HSP), Neuroendocrine Tumors (NET), Pancreatic Neuroendocrine Tumors (PanNET), Gastroenteropancreatic Neuroendocrine Tumors (GEP-NET), targeted therapy, biomarkers

## Abstract

Neuroendocrine tumors (NETs) are a diverse group of rare but clinically important neoplasms with increasing incidence and high biological heterogeneity. Heat shock proteins (HSPs) play a key role in the cellular response to stress, participating in the maintenance of proteome homeostasis and in the regulation of tumorigenesis processes such as proliferation, migration, apoptosis and development of resistance to therapy. This review analyzes the importance of HSPs in the diagnosis, prognosis and therapy of neuroendocrine tumors, as potential prognostic markers and targets of molecularly targeted therapy. The possibility of using HSP activity modulation to increase the efficacy of treatment, especially in cases refractory to standard therapeutic regimens, is emphasized. Due to the increasing clinical importance of NETs and limited therapeutic options in advanced forms, further exploration of the role of HSPs as biomarkers and therapeutic targets in this group of tumors is necessary.

## 1. Introduction

Heat shock proteins (HSPs) are a group of highly conserved proteins, constituting about 5–10% of the cell proteome under physiological conditions. Their main function is to maintain proteostasis, especially in response to environmental stressors such as elevated temperature, hypoxia or oxidative stress [Table 1] [Figure 1]. They are classified based on molecular weight into large proteins, i.e., HSP90, HSP70, HSP60, HSP40 and small HSPs [1,2].

They were first described in 1962 by Ferruccio Ritossa in the fruit fly (Drosophila melanogaster) as proteins induced by an increase in temperature [16]. They are common in pro- and eukaryotes, mainly in intracellular locations, and their synthesis is dependent on the translational activity of the cell [1,17]. In stress conditions, the MAPK pathway is activated, leading to phosphorylation and trimerization of the transcription factor HSF1. Active HSF1 binds to HSE sequences in the promoters of HSP genes, inducing their expression [18,19].

In response to cellular stress, heat shock proteins (HSPs) play a key role in maintaining proteome homeostasis [Figure 2]. They participate in protein folding, transport, degradation and refolding, as well as in the breakdown of protein aggregates [1,20]. In addition, they participate in intracellular signaling and regulation of apoptosis [1,21]. Some HSPs can localize to the cell surface, influencing antigen presentation and modulation of the immune response. They also stimulate the secretion of cytokines (TNF-α, IL-1, IL-6, IL-12, IL-10), nitric oxide (NO) and chemokines by immune cells [22,23,24,25,26,27].

After the cellular stress is relieved, HSP expression is silenced in a feedback mechanism—newly formed HSPs and HSBP1 (heat shock factor binding protein 1) inhibit HSF1 activity, limiting further transcription [28,29]. They perform various functions as foldases, holdases, sequestrases, aggregases and disaggregases—both in physiological and pathological conditions [1,30]. Both deficiency and overexpression of HSPs can disrupt cell function. Deregulation of HSP expression is observed in many cancers, including pancreatic cancer. These proteins promote proliferation, invasion, metastasis, inhibit apoptosis and support the development of resistance to anticancer therapy [31,32,33,34] [Figure 3].

Neuroendocrine neoplasms (NENs) are a heterogeneous group of neoplasms derived from cells of the neuroendocrine system, exhibiting both secretory and neuronal features. Their most common location is the gastrointestinal tract and the lungs, which are the second-most common primary location [35,36,37,38,39]. NENs can be divided into two main types: well-differentiated neuroendocrine tumors (NETs) composed of cells with oval and round nuclei and granular chromatin characterized by a slow clinical course, and poorly differentiated neuroendocrine carcinomas (NECs) with a lower degree of expression of neuroendocrine markers and high biological aggressiveness [35,36,37]. Tumors may exhibit a secretory function or not, which affects the presence of symptoms related to endocrine function and earlier diagnosis. Diagnosis of non-secretory neoplasms is usually delayed due to their long asymptomatic course [35,39].

Due to the increasing incidence and limited treatment options in advanced stages of the disease, the search for new biomarkers and therapeutic targets remains important.

## 2. The Aim of the Study

The aim of this paper is to analyze the role of heat shock proteins in the context of diagnosis, prognosis and treatment of neuroendocrine tumors of the digestive tract, lung carcinoids (NET G1–G3) and small cell lung cancer (SCLC), on their potential use as biomarkers and therapeutic targets. The paper discusses the current state of knowledge regarding the importance of individual HSP families in cellular homeostasis and carcinogenesis. Both experimental data and clinical observations are included, emphasizing the molecular mechanisms of HSP action and their involvement in disease progression, treatment response and therapeutic resistance. Therapeutic strategies aimed at HSP inhibition and the possibilities of their use in combined therapy and tumor imaging are also analyzed.

## 3. The Significance of HSPs in the Diagnosis and Therapy of Well-Differentiated Neuroendocrine Tumors of the Lung and Small Cell Lung Cancer

### 3.1. Typical and Atypical Carcinoid Tumors

Pulmonary neuroendocrine tumors (Lu-NETs) are divided into typical and atypical carcinoids, large cell neuroendocrine carcinomas, and small cell lung carcinomas [40]. Well-differentiated pulmonary neuroendocrine tumors include typical carcinoids (TC) showing <2 mitoses per 2 mm^2^ and characterized by the absence of necrosis, and atypical carcinoids (AC) showing 2–10 mitoses and/or pinpoint foci of necrosis. The course of these tumors is slow, the first symptoms are, similarly to lung cancer, dry cough, hemoptysis, and dyspnea. Carcinoid syndrome, caused by increased concentration of 5-hydroxyindoleacetic acid (5-HIAA) with hot flashes, diarrhea, or asthma, occurs with a frequency of 2–12%. Lung carcinoids constitute about 2% of all lung tumors [40,41]. Typical carcinoid occurs much more frequently (2% vs. 0.2%). The distinction between TC and AC is an important prognostic factor, the 5-year survival time is significantly shorter in patients diagnosed with atypical carcinoids [41].

The study by Niinimäki et al. showed that HSP90AB1 expression is significantly higher in disseminated lung carcinoids. Moreover, increased HSP90 expression is associated with shorter survival (*p* = 0.009) and an increased risk of cancer-related death. Based on the study, it was hypothesized that these proteins may be a tool in making therapeutic decisions in the future [42].

An attempt to use HSP90 in the therapy of neuroendocrine tumors, including those originating from the respiratory system, was the study by Zitzmann et al., in which HSP90 inhibitors AUY922 and HSP990 showed an antiproliferative effect [43].

### 3.2. Small Cell Lung Cancer

Although small cell lung cancer cells (SCLC) differ biologically from lung carcinoids, research on heat shock proteins (HSPs) has also been presented for this subtype, as they have neuroendocrine cell characteristics and a common origin. Small cell lung cancer accounts for 15–20% of lung cancers and has the most aggressive course. The median survival is approximately 12 months, and the poor prognosis is due to rapid tumor growth, early metastasis, and frequent resistance to initially effective treatment [44].

#### 3.2.1. The Role of HSP70 and HSP90 in the Proliferation, Invasiveness and Chemo-Resistance of SCLCs

The association of heat shock proteins with resistance to treatment is provided by a study by Du and co-authors, in which the activity level of HSP90α was determined by Enzyme-linked Immunosorbent Assay (ELISA) in tumor cells treated with doxorubicin (DOX) or ABT-737 in mouse xenograft models. Both therapies led to an increase in extracellular levels of HSP90α, which activated the AKT/β-catenin signaling pathway and inhibited glycogen synthase kinase 3β (GSK3β). This resulted in tumor growth and inhibited apoptotic mechanisms, thereby increasing resistance to treatment. At the same time, administration of the monoclonal antibody anty-HSP90α inhibited tumor growth and enhanced apoptosis of its cells, suggesting that it may be a potential candidate for combination therapy with DOX in the treatment of SCLC [45].

#### 3.2.2. HSP70 as a Diagnostic and Prognostic Marker for SCLC

Heat shock proteins may also be potential biomarkers of SCLC. An analysis by Balázs and co-authors showed that serum levels of HSP70 in 70 SCLC patients were significantly higher than among an age-matched healthy population. Moreover, the highest HSP70 levels were measured in patients with stage IV advanced SCLC, which, combined with a 3-fold higher 1-year risk of death in patients with high initial HSP70 levels (>2.42 ng/mL), suggests that this protein could become a potential diagnostic and prognostic marker of this cancer [46].

#### 3.2.3. The Potential Utility of Heat Shock Proteins

Although heat shock protein inhibitors are not the mainstay of SCLC treatment due to their limited activity as monotherapy in clinical trials, they may be useful in combination therapy, especially in cancers resistant to conventional treatment. One potential candidate for combination treatment of SCLC is the HSP90 inhibitor NVP-AUY922, which in a study by Yang and co-authors synergized with the BCL-2 inhibitor ABT-737 to induce apoptosis of tumor cells with BCL-2 overexpression better than the individual drugs in monotherapy. A suggested mechanism for NVP-AUY922 antitumor activity is blocking pNF-κB activity and down-regulation of AKT and ERK activating MCL-1 expression, which in turn results in the development of resistance against ABT-737 [47].

A promising drug that could help in the therapeutic management of chemotherapy-resistant SCLC is also the HSP90 inhibitor ganetespib. However, Subramaniam and colleagues, in an expert analysis, highlight that this drug has not shown significant clinical benefit in combination with docetaxel in the treatment of non-small cell lung cancer (phase IIb GALAXY-1 and phase III GALAXY-2 trials), as it has in SCLC monotherapy (phase II trial) [48,49,50]. A phase Ib/II clinical trial conducted by Subramaniam and co-authors on 11 patients (including 9 in phase Ib and 2 in phase II) showed that the recommended dose of the drug in patients with advanced solid tumors is 150 mg/m^2^ on days 1 and 8, with simultaneous administration of doxorubicin (50 mg/m^2^) on day 1, in a 21-day cycle. In the study, the maximum tolerated dose (MTD) was not reached, and side effects included diarrhea, nausea, fatigue and elevated transaminase levels. The objective response rate (ORR) of the combination treatment was 40%, indicating that it makes sense to conduct further studies on the synergistic effects of the two drugs [51]. However, the study was discontinued by sponsor probably due to the failure of the GALAXY-2 clinical trial. Nevertheless, ganetespib remains a promising chemotherapeutic agent for the combination treatment of SCLC [48,49,50,51].

Due to the side effects of the drugs, STA-8666 which is a combination of the HSP90 inhibitor and the active metabolite of irinotecan SN-38, was developed and started clinic studies. Gaponova at all report that the drug was well tolerated and resulted in more stable and rapid cell cycle arrest and enhanced apoptosis of SCLC cells in vitro and in vivo in a PDX (patient-derived xenografts) model than irinotecan monotherapy [52].

### 3.3. Neuroendocrine Tumors of the Gastrointestinal Tract (GEP-NET) and Pancreas (Pan-NET)

Neuroendocrine tumors of the digestive system are rare, usually slow-growing neoplasms with characteristic histological and clinical features showing high heterogeneity. In recent years, a gradual increase in the number of new cases has been noted [53,54]. Neuroendocrine carcinomas (NEC) with high clinical aggressiveness are rare in the digestive tract, and scientific evidence on optimal therapeutic strategies is limited [55]. The small intestine is the most common primary location of NETs. Carcinoid syndrome usually occurs when neuroendocrine tumors of the jejunum and ileum metastasize to the liver. About 20% of neuroendocrine tumors are associated with hereditary genetic syndromes, i.e., multiple endocrine neoplasia type 1 (MEN1) or neurofibromatosis type 1 (NF-1) [53,54]. Pancreatic neuroendocrine tumors (P-NET/Pan-NET) are a heterogeneous group of tumors, accounting for 1% to 2% of pancreatic tumors, and their incidence is increasing. pNETs are clinically classified as secreting or nonsecreting, depending on whether they release hormones such as insulin, gastrin, vasoactive intestinal peptide (VIP), glucagon, somatostatin, or serotonin. Nonsecreting pNETs, occurring more frequently, are diagnosed at later stages due to their long asymptomatic course [56]. The stage of advancement and histopathological classification are essential for making therapeutic decisions. The prognosis of patients with metastatic PanNETs is heterogeneous and very diverse. New specific biomarkers are sought that will allow for earlier diagnosis and selection of optimal therapy [57,58].

In the study by Gamboa et al., it has been shown that 34% of primary neuroendocrine tumors of the stomach, intestines, and pancreas overexpress HSP90, which translates into a higher risk of relapse. In patients with liver metastases, high HSP90 levels correlated with a shorter 3-year progression-free survival (25% vs. 49%). The role of HSP90 in the pathogenesis of GEP-NETs has made its inhibitors an attractive therapeutic strategy [59].

In the study by Lundsten et al., the effect of the HSP90 inhibitor onalespib was assessed on NET cell lines, including BON obtained from lymph node metastasis of pancreatic carcinoid tumors. The concept of radiosensitization was used, i.e., the potential effect on the ability of tumor cells to respond to radiation and thus increase the efficacy of therapy, e.g., by influencing the mechanisms of DNA damage and repair. Onalespib was shown to reduce cell viability and spheroid growth. Moreover, the combination of onalespib and 177 Lu-DOTATATE exerted synergistic therapeutic effects. Western blot analysis showed a decrease in the expression of the epidermal growth factor receptor (EGFR) and an increase in the expression of the histone family member γ H2A X (γH2AX), a sensitive marker of DNA damage, after combined treatment with onalespib and 177 Lu-DOTATATE [60].

In another study by Lundsten et al. conducted in vivo on mice with NET xenografts (QGP-1 and BON lines), the previous observations were confirmed. Onalespib in monotherapy inhibited tumor growth, but the greatest therapeutic effect was observed when combined with 177Lu-DOTATATE. In mice treated with combination therapy, a significantly higher rate of complete remission was noted, 29% in the case of combined treatment vs. 8% in the 177Lu-DOTATATE group. Combination treatment was well tolerated, without significant systemic toxicity or even potential nephroprotection of onalesib [61].

Based on these results, onalespib seems to be an effective and safe radiosensitizer that can increase the effectiveness of radioisotope therapy by inhibiting HSP90-dependent protective mechanisms [Table 2].

In vitro studies have confirmed the efficacy of HSP90 inhibitors, i.e., AUY922, HSP990, IPI-504 and 17-AAG, on gastrointestinal neuroendocrine tumor cells. The mechanism of action of AUY922 and HSP990 was based on the inhibition of cell viability through the induction of apoptosis and the reduction in the expression of neuroendocrine receptors ErbB and IGF-I, reduced phosphorylation of Erk and Akt and the induction of HSP70 expression [43,62]. The potential activity of IPI-504 has been demonstrated in human insulinoma and pancreatic carcinoid cells. IPI-504 has antiproliferative effects through the reduction in IGF-1 and PI3K/Akt/mTOR expression. Combination of IPI-504 with mTOR or Akt inhibitors also resulted in enhanced antiproliferative activity [62]. The Hsp90 inhibitor 17-(allylamino)-17-demethoxygeldanamycin (17-AAG) showed the ability to inhibit proliferation in NET cell lines and induced the loss of EGFR, IGF1R and VEGFR2 [63,64].

Hofving et al. used the GOT1 cell line obtained from a liver metastasis of a midgut carcinoid tumor and the P-STS cell line obtained from a primary tumor, a NET of the terminal ileum, to evaluate the response of small intestinal neuroendocrine tumor (SINET) cells to radionuclide therapy with 177Lu-octreotate. It was observed that the administration of HSP90 inhibitors during therapy showed synergistic antitumor activity. The study distinguished ganetespib, which not only improved the efficacy of 177Lu-octreotate therapy in vivo, but also its enhancing effect was maintained ex vivo in SINET cells [65].

### 3.4. The Use of Metformin in the Treatment of P-NETs and Its Interactions with Heat Shock Proteins

Studies conducted by Vitali et al. have shown that metformin—a well-known antidiabetic drug with documented anticancer activity—also exhibits antiproliferative activity against pancreatic neuroendocrine tumor cells by reducing the concentration of HSP70. This effect is associated with the activation of the AIP (aryl hydrocarbon receptor-interacting protein)-dependent pathway, which plays a key role in regulating the cellular response to environmental and stress factors. Under the influence of metformin, the concentration of HSP70 is reduced, which weakens the protective mechanisms of cancer cells and facilitates the induction of apoptosis, increased expression of AIP, which interacts with the aromatic hydrocarbon receptor (AhR) and activation of the transcription factor Zac1 (zinc-finger protein regulating apoptosis and cell cycle), which is an effector of AIP. Enhancement of the AIP–Zac1–AhR pathway in response to metformin leads to inhibition of phosphorylation and activation of the mTOR pathway and to the intensification of apoptotic mechanisms in PNET cells and limitation of their proliferative capacity. The above observations indicate that metformin may also be an effective therapeutic tool in the treatment of pancreatic neuroendocrine tumors. Its effect on the level of HSP70 and elements of the AIP pathway suggests the possibility of including this drug as a component of combined therapy, especially in cases refractory to standard treatment [57] [Figure 4].

### 3.5. Application of HSP90 in Positron Emission Tomography (PET)

Pancreatic neuroendocrine tumors are characterized by increased expression of HSP70 and HSP90. These proteins are therefore a potential target in the diagnosis and treatment of Pan-NETs. In the study by Nordemann et al., an attempt was made to improve the diagnosis and treatment by positron emission tomography (PET). A fluorine-18-labeled HSP90 ligand was proposed, which was then tested in vitro and in a mouse model. A high affinity of the ligand for heat shock protein was demonstrated in vitro, but in vivo it was rapidly metabolized and excreted by the mouse body [66].

## 4. Conclusions

This literature review comprehensively summarizes the current state of knowledge on the role of heat shock proteins (HSPs) in the diagnosis and treatment of neuroendocrine tumors (NETs), with particular emphasis on their potential as biomarkers and components of combined treatment for tumors resistant to standard chemotherapy. As demonstrated in a study by Gamboa et al. patients who underwent radical resection of primary, non-metastatic NETs, a high level of HSP90 was associated with a shorter time free to recurrence. In contrast, patients with liver metastases with high HSP90 levels correlated with worse 1- and 3-year progression-free survival compared to patients with low HSP90 levels [59]. Similarly, in the case of lung carcinomas, a study by Niinimäki et al. showed that elevated HSP90 expression is associated with shorter survival [42]. Thus, HSP90 levels may become a useful prognostic marker in the future to select patients at high risk of recurrence. Heat shock proteins also represent a promising therapeutic target for the treatment of NET. Although clinical trials have not demonstrated the high efficacy of HSP90 inhibitors in monotherapy, they are a potential adjunct to combination therapy for refractory cancers resistant to conventional therapy [62,63,64]. Moreover, onalespib, improves the efficacy of radiotherapy for GEP-NET, including SINET in combination with radioisotope therapy. The study by Lundsten et al. demonstrated the potential for a significant increase in complete remission rates with combination therapy, and even the potential nephroprotective effect of onalesib, which may broaden the pool of candidates eligible for this therapy [61]. Importantly, metformin, too, may be a potential component of NET therapy using its effect on heat shock proteins—studies show that the compound inhibits PNET proliferation by enhancing the AIP-Zac1-AhR pathway and downregulating HSP70 expression [66]. The progress made so far in the therapy of neuroendocrine tumors using HSP inhibitors is promising, many preclinical studies require validation. However, the analyses to date suggest that heat shock proteins play an important role in the pathophysiology of many NETs, and can serve not only as diagnostic and predictive biomarkers, but above all as a pillar of anticancer therapy. Therefore, further exploration of the topic is a promising direction for oncological research.

## Figures and Tables

**Figure 1 cells-14-01501-f001:**
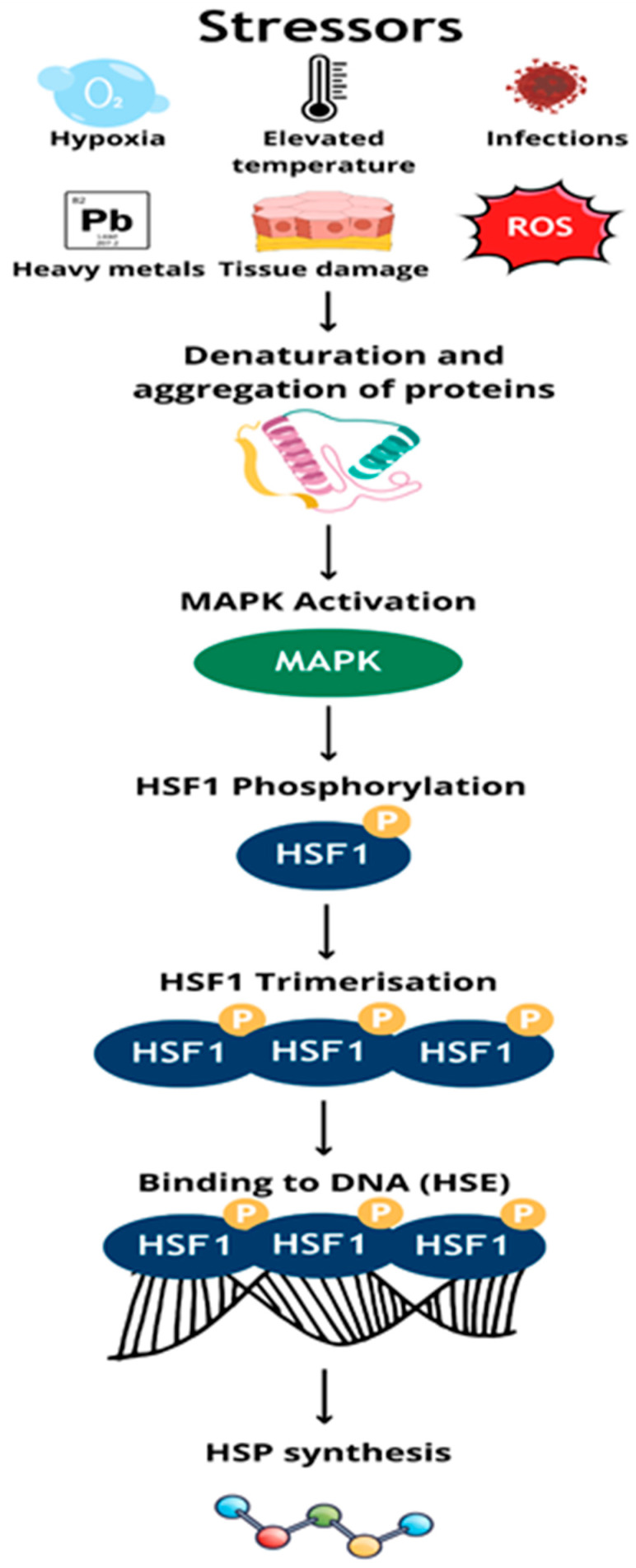
The influence of stress stimuli on the synthesis of heat shock proteins.

**Figure 2 cells-14-01501-f002:**
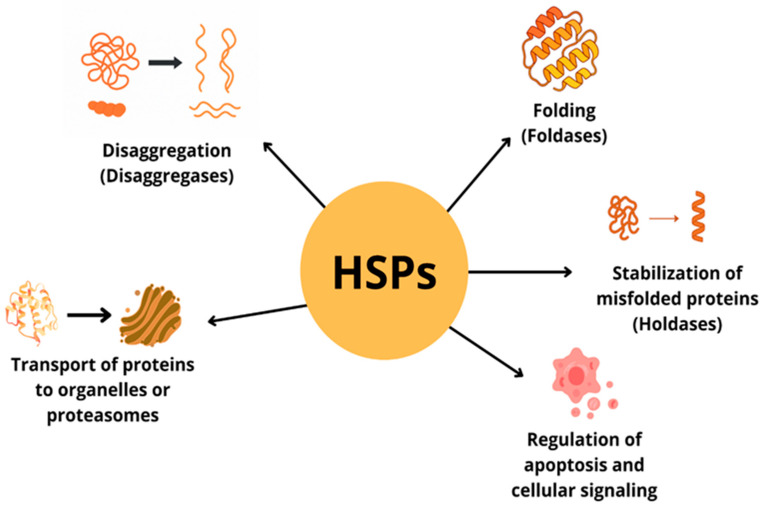
Participation of heat shock proteins in molecular physiological processes.

**Figure 3 cells-14-01501-f003:**
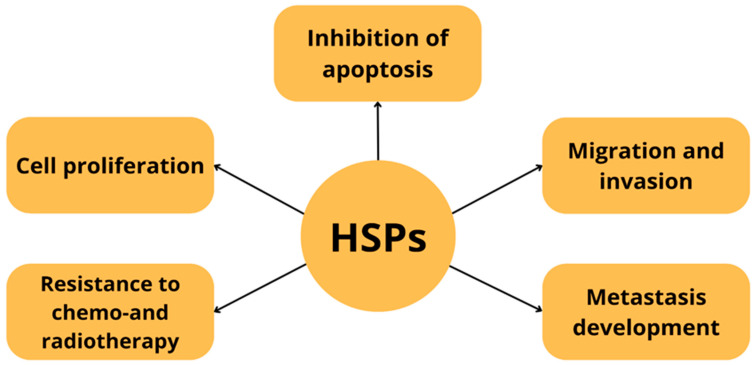
The involvement of heat shock proteins in neoplastic processes.

**Figure 4 cells-14-01501-f004:**
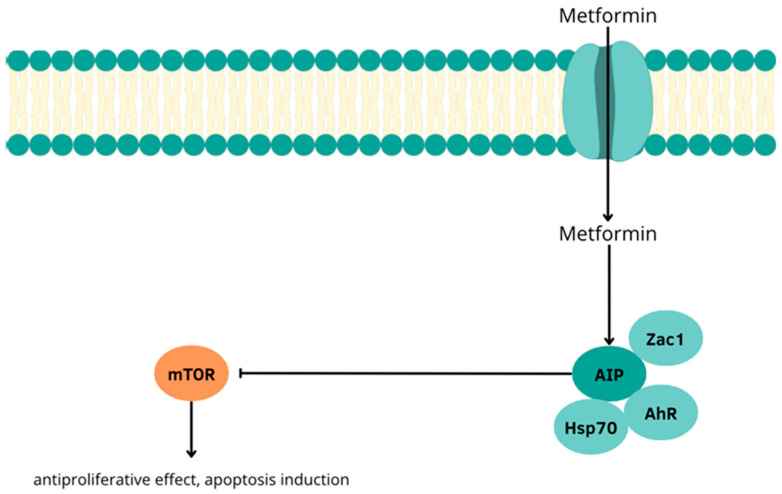
Potential mechanism of mTOR phosphorylation inhibition by metformin in pancreatic neuroendocrine tumor cells.

**Table 1 cells-14-01501-t001:** The role of selected families of heat shock proteins in the regulation of cancer processes.

Protein Family Name	Function Performed	Role in Carcinogenesis
HSP20	Hsp20 is expressed in many tissues, but is most abundant in muscle including heart muscle In the heart, it protects against ischemia-induced damage and β-agonist-dependent remodeling, as well as apoptosis. In addition, it inhibits aggregation and activation of human platelets outside the cell. It is an inhibitor of apoptosis [3,4].	Expression of the HSP20 protein can inhibit HCC cell growth by both reducing cell proliferation signals and activating the apoptosis pathway [5].
HSP40 (family DnaJ)	It is the largest and most diverse subgroup of the HSP family. The Hsp40 protein acts in specific pairing with the Hsp70 protein to regulate the ATPase activity of HSP70. It is involved in promoting protein folding, translocation and degradation [1,6].	The HSP40 family plays an important role in the development, progression, metastasis and chemoresistance of various malignancies [7]. Overexpression of the HSP40 family has been demonstrated in various cancers including colorectal, breast, prostate, ovarian, liver, head and neck cancers. HSP40 acts not only in tandem with HSP70 but also as an indirect regulator of the HSP90 complex, with roles in both pro- and anti-tumor processes [7].
HSP60 (chaperonin\Cpn60)	Localized mainly in mitochondria (80–85%), HSP60 is responsible for ATP-dependent protein folding. In addition, it is involved in the refolding and degradation of mitochondrial proteins. The cytoplasmic form of HSP60 is involved in cell signaling in various cell types, such as cardiomyocytes and hepatocytes [8].	HSP60 is overexpressed in cancers such as colorectal cancer, non-small cell lung cancer (NSCLC), breast cancer and hepatocellular carcinoma (HCC), while at the same time HSP60 levels are lower in bladder cancer and clear cell renal cell carcinoma [8,9,10]. HSP60 has been linked to cancer pathogenesis and progression at the level of various mechanisms. The cytoplasmic form of HSP60 is involved in cell signaling in various cell types, such as cardiomyocytes and hepatocytes [8].
HSP70	Responsible for the folding of non-native proteins, prevents their aggregation [11]. Controls the quality of misfolded proteins and is responsible for co-translational and post-translational folding of de novo nascent proteins [11].	Stimulates both innate and adaptive immune response expression correlates with better prognosis in gastric and colorectal cancer and poor prognosis in lower rectal and squamous cell carcinoma [12].
HSP90	Key regulator of proteostasis under both physiological and stress conditions in eukaryotic cells. Participates in DNA repair, immune response. It has a large number of co-chaperones [13].	Associated with tumor cell invasion and migration. By increasing transcription and expression of vascular endothelial growth factor receptor (VEGFR), it is involved in angiogenesis of cell proliferation, migration and invasion. The expression level of HSP90 has been recognized as a potential biomarker of poor prognosis in lung cancer or esophageal cancer, gastrointestinal neuroendocrine tumors [9].
HSP27	ATP-independent, small molecule chaperone. Hsp27 phosphorylation is involved in cell migration, modulation of cell cycle progression through inhibition of the MEK/ERK signaling pathway or interaction with p53 and inhibition of apoptosis [14].	Overexpressed in many cancer cell types Increased Hsp27 expression or phosphorylation is associated with chemotherapeutic resistance, tumor progression and metastatic potential. Associated with poor prognosis in gastric, liver, prostate cancer and osteosarcoma [2,15].

**Table 2 cells-14-01501-t002:** A review of selected studies on the therapeutic use of heat shock proteins in the treatment of neuroendocrine tumors.

Authors of the Study	Trial Model	Results
Niinimäki J et al. [42]	Patients with pulmonary carcinoid tumor	HSP90AB1 expression was significantly elevated in metastatic PC tumors (*p* < 0.0001) HSP90 protein expression was associated with shorter disease-specific survival (DSS) (*p* = 0.009) and increased risk of disease-specific death
Zitzmann et al. [43]	Neuroendocrine tumor cells of pancreatic (BON1), midgut (GOT1) and bronchopulmonary origin (NCI-H727).	The HSP90 inhibitors AUY922 and HSP990 inhibited the viability of BON1 cells, NCI-H727 and GOT1 cells. Inhibition of HSP90 induces apoptosis in neuroendocrine tumor cells. Inhibition of HSP90 in human BON1 pancreatic cells was associated with a significant increase in the number of cells in the G2/M phase, while no effect on cell cycle distribution was observed in NCI-H727 and GOT1 cells.
A.C. Gamboa et al. [59]	Patients with non-metastatic GEP NETs and GEP NETs with liver metastases.	Among patients who underwent resection (R0 or R1) due to primary, non-metastatic GEP-NET patients with high HSP90 expression had lower 1- and 3-year survival rates compared to patients with low HSP90 expression. High HSP90 expression is associated with poorer recurrence-free survival. Patients with high HSP90 expression and liver metastases had lower 1- and 3-year survival rates compared to patients with low HSP90 expression.
Lundsten et al. [60]	NET cell lines BON (cells from lymph node metastasis of a carcinoid tumor of the pancreas), NCI-H727 (neuroendocrine cell line derived from a human lung carcinoidand), NCI-H460 (large cell lung carcinoma human cell line with neuroendocrine features).	Onalespib was able to synergistically enhance the treatment of 177 Lu-DOTATATE in a manner specific to SSTR. The mechanisms of radiosensitivity of onalespib included a reduction in EGFR expression and the induction of apoptosis.
Lundsten S et al. [61]	NET cell lines BON, established from a lymph node metastasis of a pancreatic carcinoid tumor Squamous cell carcinoma cell line UM-SCC-74B	Potentiation of 177 Lu-DOTATATE in a neuroendocrine tumor xenograft model, resulting in delayed tumor growth, increased total remission rates, and reduced renal toxicity.

## Data Availability

No new data were created or analyzed in this study.

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
