# Peer review of "Heat Shock Proteins in Gastrointestinal and Lung Neuroendocrine Neoplasm: Diagnostic and Therapeutic Perspectives"

_cells, 2025, doi:10.3390/cells14191501_

Round 1
Reviewer 1 Report
Comments and Suggestions for Authors
Challenging paper to review. This is more basic science showing the potential for HSP in diagnosis and potential therapy but it is still very premature and there is a paucity of clinical data in humans. The data presented and reviewed are in spheroids and animal models.
It would seem to review the background to HSP well but goes downhill from a clinical perspective thereafter as most of the data is from animal or cell models.
I do not have sufficient expertise to comment on some of the basic science included here. It seems well written and the English language is good
As a consequence i think it is of limited interest to the general reader. If the target audience is a clinical scientist then it is a reasonable review but if targeted at clinicians I have reservations.
There are repeated references to small cell lung cancer which is not a NET.
Line 70 should read tumours not neoplasms. NENS are divided into NETs and NECs.
Line 182, 20% of NENs associated with hereditary genetic syndromes is clearly wrong. It is very infrequent.
Interestingly none of the authors appears to be NET specialists. I wonder whether this paper may be better targeted at other journals
Author Response
Dear reviewer, we are extremely grateful for your review.
Challenging paper to review. This is more basic science showing the potential for HSP in diagnosis and potential therapy but it is still very premature and there is a paucity of clinical data in humans. The data presented and reviewed are in spheroids and animal models.
It is true that the current state of knowledge regarding the importance of heat shock proteins and their diagnostic and therapeutic potential in cancers as rare as neuroendocrine tumors is limited. The goal of our work is to summarize current scientific achievements and inspire the development of further studies that could broaden our current perspective on these molecules. Given that individual therapies bring significant clinical benefits to patients diagnosed with neuroendocrine tumors, we believe this topic is important and worthy of the reader's attention.
It would seem to review the background to HSP well but goes downhill from a clinical perspective thereafter as most of the data is from animal or cell models.
The current state of knowledge regarding the importance of HSPs in the etiology and therapy of NETs is undoubtedly based on cellular and animal models, but the results of the studies conducted offer the opportunity for further research in animal models. However, there are also studies that use prognostic significance in a population of patients with NET.
I do not have sufficient expertise to comment on some of the basic science included here. It seems well written and the English language is good
As a consequence i think it is of limited interest to the general reader. If the target audience is a clinical scientist then it is a reasonable review but if targeted at clinicians I have reservations.
We treat patients diagnosed with neuroendocrine tumors daily while also conducting research. We hope that this work will reach both scientists and physicians who, like us, strive to expand knowledge about potential therapies for this narrow group of patients.
There are repeated references to small cell lung cancer which is not a NET.
Thank you for your attention. We have explained in the manuscript why we also described this type of cancer.
Line 70 should read "tumors," not "neoplasms." NENS are divided into NETs and NECs.
Thank you for your attention, we have made a correction.
Line 182, "20% of neuroendocrine tumors (NENs) associated with hereditary genetic syndromes," is clearly incorrect. It is very rare.
Below we have provided references confirming our citations. However, we are fully aware that the percentage of genetically determined NETs is closely related to the type/origin of the tumour, hence there may be significant discrepancies in the literature.
Crona J, Skogseid B. GEP- NETS UPDATE: Genetics of neuroendocrine tumors. Eur J Endocrinol. 2016 Jun;174(6):R275-90. doi: 10.1530/EJE-15-0972. PMID: 27165966.
Wang R, Zheng-Pywell R, Chen HA, Bibb JA, Chen H, Rose JB. Management of Gastrointestinal Neuroendocrine Tumors. Clin Med Insights Endocrinol Diabetes. 2019 Oct 24;12:1179551419884058. doi: 10.1177/1179551419884058. PMID: 31695546; PMCID: PMC6820165.
Interestingly, none of the authors appear to be specialists in the field of NETs. I wonder if this article might be better published in other journals.
The clinical unit where we work and conduct our daily research includes an Endocrinology Clinic which has been certified as a 'Centre of Excellence' by the European Neuroendocrine Tumour Society (ENETS). Consequently, we treat patients diagnosed with neuroendocrine tumours on a daily basis, while also monitoring our progress and conducting our own research with this patient group.
Reviewer 2 Report
Comments and Suggestions for Authors
The topic is cutting-edge and deserves to be covered with this review. The authors should improve the quality of the figures (they appear a little blurred).
The bibliography doesn't seem to fulfill the requirements of the journal. Moreover, the title of the reference is the world "Literatura" that of course is not English!
The authors should comment on the best diagnostic techniques to get good samples for these molecular analyses.. In this regard, cite the relevant paper PMID: 35915956
A table with the main studies supporting the authors findings would be helpful to the readership
Author Response
Thank you for your comments. We have revised the literature section. We will send all graphs attached to the article in separate files to the editor in order to optimise their quality.
As suggested, we have included a table summarising the most important studies on NET and HSP in the article.
Round 2
Reviewer 1 Report
Comments and Suggestions for Authors
The changes make little difference in my opinion. It depends on the target audience and journal editorial policy
I doubt if it will be of interest to clinicians as mainly basic science with minimal relevant clinical relevance
On the other hand if for a basic science audience it seems satisfactory but is just a review rather than showing anything original
Reviewer 2 Report
Comments and Suggestions for Authors
The manuscript is OK